# Psychosocial Support within the Context of Perinatal Palliative Care: The “SORROWFUL” Model [note 1]

**DOI:** 10.3390/children10010038

**Published:** 2022-12-25

**Authors:** Kerstin von der Hude, Lars Garten

**Affiliations:** Department of Neonatology, Charité—Universitätsmedizin Berlin, 13353 Berlin, Germany

**Keywords:** neonates, bereaved parents, life-limiting disease, dying, birth

## Abstract

Against the background of a society that tends to underrate the grief experienced by parents whose infants have died prematurely, the model “SORROWFUL” is presented here with the intent to highlight the significance of the death of a newborn for the affected family. It is a supportive tool in counseling for parents grieving the (impending) loss of an infant(s) during peri- or neonatal life and may be implemented within the parental psychosocial support setting beginning with the initial diagnosis until well after the death of the child. The model intentionally allows flexibility for cultural and individual adaptation, for the accommodation to the varying needs of the affected parents, as well as to available local resources.

## 1. Introduction

### 1.1. The Special Situation for Parents Suffering the Premature Loss of an Infant

Parents whose children die close to birth have very little time to come to know their children and establish a bond. The grieving process not uncommonly begins in advance of the actual moment of death, potentially when the pre- or postnatal diagnosis of a life-incompatible condition had been made [1], with a premature birth, or after the subsequent birth of the baby. From the moment a grim prognosis is presented, parents are confronted with a plethora of conflicting feelings. Along with joy comes sadness and grief; hope is admixed with worry, fear, and resignation. Loving feelings for the child struggle with instincts for emotional self-preservation in anticipation of a fatal outcome.

Many parents may feel guilty about their situation, feel responsible, or feel incompetent [2,3]. They may suffer from feelings of powerlessness and helplessness stemming from their inability to help or protect their child. Given the turmoil within and all around them, many parents find it difficult to connect emotionally to their sick or dying child after birth. 

Parents may hesitate at first to get close to their child either physically or emotionally, and many find it difficult to allow an intimate relationship to develop. Parents may repeatedly attempt to maintain a “safe” emotional distance from their children to shield themselves. They subscribe to a widespread perception that the loss of a newborn infant may be more bearable if a strong emotional bond has not been established beforehand. Studies, however, suggest that the more clearly defined the bond is to the deceased, the less complicated the grieving process. An ambivalent bond shared with someone close who is deceased may be considered a risk factor complicating the grieving course and may, not uncommonly, lead to illnesses such as depression [4].

Afflicted parents are faced with the additional challenge that their grief after their baby’s death receives a more limited acknowledgment by the people around them and society in general. To a large degree, this is because the child had not yet established itself within the family or its community. Few persons could have gotten to know the child, and consequently, only a few can know for whom it is that the parents grieve.

In this lies the danger that parents grieving the loss of their infant may feel socially and emotionally isolated from those around them who seem to have moved on more quickly after the child’s death. The child’s oftentimes brief lifetime, possibly spent only in the delivery room or the intensive care unit, makes it difficult to imagine that it found acceptance as a full member of its family. Therefore, the space left empty in the parents’ lives is barely perceptible to the family and people around them. When only the parents have known the child, no one other than the parents will miss him.

The unease felt by many in dealing with the thought of infant death and dying, as well as the lack of a personal connection with the deceased child, may contribute to the death seldom being mentioned and for the parents’ grief to appear to others as being overwrought and out of proportion.


***Quotation of a parent**: “Shortly after my baby had died hardly anyone one spoke with me about it any longer. This made me feel as if it had died a second time”*


Among the observations noted in a Danish epidemiological study was that the loss of a small child is not comparable to the loss of something small. In the investigation by Li et al. (2003) [5], data from all children who had died in Denmark between 1980 and 1996 were tabulated and cross-referenced with health data from their parents. Of the total group of 12,072 deceased children, newborns accounted for 37%. Their analysis revealed two significant points:The risk of death for bereaved mothers during the three-year period after their child’s death was 40% greater than for age-matched peers who had not lost a child.This increase in maternal death risk was independent of the age of the child at its death.

The study concluded that this observed average 40% increase in mortality risk in mothers was indicative of a newborn child’s death weighing just as heavily as the death of an older child. 

### 1.2. Bereavement Support in Perinatology

Maternity and neonatal (intensive) care unit personnel are often the only persons, aside from the parents, who have had the chance to have known the baby while alive, and among the few persons with whom the parents can share memories. They stand as witnesses to the fact that the child did once exist. When the child has spent its oftentimes short life entirely within a neonatal (intensive) care unit, the parents frequently come to see the team as a surrogate family and may suffer an additional sense of loss, saying farewell to the staff after the baby’s death. 

The return to everyday life—childless—is often marked by an absent or inadequate network providing individual support for the parents during their ongoing grieving process. Given the exceptional circumstances facing grieving parents, the authors would maintain that the responsibility of a clinic to provide adequate care to infants and parents does not end when the inpatient stay is concluded. More importantly, it is necessary, even after the death of the child, to offer sufficient support for the family as they reenter everyday life and to provide or initiate a steadfast presence as they make their transition from hospital to home [6,7]. 

The aim of the “SORROWFUL” model presented in this paper is twofold: to provide clinically relevant information for multidisciplinary care teams and counselors tasked with supporting (expectant) parents experiencing exceptional life situations, as well as to offer practical guidance in the efficacious provision of psychological support. The model addresses a time span covering before, during, and after the child’s death. It does not claim to provide answers to every question. It attempts rather to orient caregivers to some important concerns that attend to the death of an infant, to focus international attention on these fundamental human issues, and to encourage a process of continual improvement in dealing with them. 

Aspects of this work have been orally presented at lectures, seminars, and workshops and a preliminary set of basic recommendations addressing psychosocial support within the context of PnPC have been previously published in German [8,9].

## 2. Methods

The ‘SORROWFUL’ model is based on the working model ‘TRAUER’ (German, Trauer = grief) developed by Kerstin Lammer [4]. It has been modified and adapted to the specific needs of families who have either lost a child or are preparing for the death of their child during the peri- and neonatal periods.

As presented here, the “SORROWFUL’ model reflects not only scientific evidence but, more significantly, an experience base of more than 25 years in providing bereavement care to families of newborns at the perinatal referral center at Charité Universitätsmedizin Berlin. Additionally, it incorporates input from the “PaluTiN Group”, an interprofessional and interdisciplinary German national expert panel responsible for developing recommendations for palliative care and grief counseling in peri- and neonatology [6]. It is applicable to the psychosocial support of parents at the time of diagnosis as well as after the death of the child. Since 2018, grief counseling for families at the perinatal referral center at Charité Universitätsmedizin Berlin has been provided in strict accordance with the specific principles of the “SORROWFUL” model.

The perinatal referral center at Charité Universitätsmedizin Berlin consists of two NICUs, one intermediate care unit, and one special care unit. At our institution, approximately 30–40 neonates with life-limiting conditions are cared for each year under the designation of palliative care. Since 2016, a special perinatal palliative care (PnPC) team has provided interdisciplinary and interprofessional prenatal palliative care consultations in collaboration with the prenatal diagnosticians, obstetricians, and midwives of our Department of Obstetrics as a part of family-centered fetal care. Our PnPC consultations are based on the principles of prenatal advance care planning that have been published in detail previously [10,11]. The hospital-based PnPC team consists of 

Neonatal intensive care nurses specialized in pediatric palliative care, working in both tertiary neonatal intensive care units or in our intermediate care unit,Two perinatal parent counselors specialized in bereavement counseling and trauma-sensitive counseling, andOne neonatologist specialized in pediatric palliative care.

In Germany, the additional costs of hospital-based PnPC services are not fully covered by the German public health care system. At our institution, additional costs for psychosocial counseling and grief counseling associated with PnPC are primarily funded by charitable foundations, as well as partially subsidized in the outpatient setting by statutory health insurance, which provides a fixed payment rate per case of slightly less than EUR 100/quarter year. 

## 3. The ‘SORROWFUL’ Model

The ‘SORROWFUL’ model for psychosocial bereavement support within the context of perinatal palliative care was developed to offer guidance to those tasked with providing individualized support to bereaved families, ensuring that such support is extended in a respectful, understanding, and open-minded manner. It can help to set therapeutic processes into motion and expand perceptions as to what options may be possible.

The core principle of the “SORROWFUL” model rests upon recognizing that optimal support is derived from actively asking the families what their individual needs and wishes are. Support needs to be offered with the utmost attentiveness to individuality, with sensitivity to culture, ethnicity, gender, and religion, and without disturbing intimacy.

### 3.1. S—Steadfast Support 

Parents need active outreach services for reliable ongoing support. At the beginning of the grieving process, it is very seldom that families take an active role in contacting available supportive services. The invitation to “feel free to call if you need anything” is seldom taken up since, at the beginning of the grieving process parents, are often still in a state of shock, unaware of what it is they need and not knowing what next steps are coming. 


**
*Quotation of a parent*
**
*: “After this conversation, when we learned how serious our baby’s situation was, we went home in a daze. At the end, the doctor had asked us if we had any questions. But we were so overwhelmed that we didn’t know what to ask. We had no idea about any of this.”*


A proactive counseling service can support the parents with guidance and information, helping them through shared discussions to organize their thoughts and identify their fears, and the moment to figure out for themselves what feels right, what first steps in this highly stressful situation they can picture themselves taking. 

Parents depend upon the open-minded cooperation of each of the inter-professional staff members involved in their case [12]. This is essential in order to be able to keep all of the parents’ needs (both medical and non-medical) in mind throughout the entire process. In these high-stress situations, parents should ideally not have to make far-reaching decisions under time pressure and without the offer of psychological support [13].

### 3.2. O—Offer Gentle and Compassionate Prompts

Gentle and compassionate prompts may assist parents in transitioning from powerlessness and disorganization into a structure of their own design. From the initial discussion of the diagnosis, the (expectant) parents find themselves in a high-stress situation. Despite the stress and excessive demands, however, they still need to make decisions for their (un-) born child while continuing to function in their various other social roles (as employees, caregivers to their other children, information providers to their families and friends, etc.) [12].


**
*Quotation of a parent*
**
*: “It felt as if I was juggling many balls at once, and that I must not allow any of them—not a single one—to fall.”*


Early offers of psychosocial counseling services can help parents think their way through their next steps and develop a structure for themselves. There are many issues—often non-medical in nature—that may be highly relevant to parents yet which they themselves will not raise. This may be in part due to their having no direct role to play in the clinical care of their sick baby, and therefore are not suitable in the medical discussions. Ongoing psychosocial discussions, however, can offer emotional relief while at the same time nurturing in the parents a growing sense of self-efficacy when they are able to develop and implement individual solutions or identify approaches to solutions.

After the death of a child, many parents initially remain under a great deal of tension and realize only gradually that no further vital decisions remain to be made on their child’s behalf. This sense of “doing nothing” feels strange and may be perceived as becomes burdensome in its own way, standing in stark contrast to the inner tensions of the previous days, weeks, or months. Emotional processing lags behind cognitive understanding, and as a result, many parents react with an impulsive need to “do something” and often seize upon the task of immediately organizing the funeral. 

The organization and design of burial arrangements are important steps in the grieving process, though often not the first. Exceptions hold true here, especially for persons whose religious faith prescribes a prompt burial. Otherwise, it is more advisable to gently intercept this initial inner impetus to act. The parents should be given the opportunity to reconcile emotionally with their new situation. Only gradually will they develop a sense of what the next steps might entail, but initially, there should be a space or a time put aside for all the feelings suppressed while needing to function.

### 3.3. R—Realizing and Comprehending Death with all Senses 

The realization and comprehension of death take time. As already noted, parents in this situation often have very limited shared time with their unborn or newborn child suffering from a life-limiting illness. They are challenged to adjust to their child’s dying yet cannot fully prepare themselves for its death. Even after the child has died, they need additional time for its reality to be felt. They understand intellectually that their child has died, but often it does not feel real. 


**
*Quotation of a parent*
**
*: “My heart can’t catch up with my brain yet. I know my little one has passed away, but I still feel like I’m dreaming and hope I’ll suddenly wake up and none of this will have happened.”*


To comprehend death with all one’s senses goes beyond mere intellectual understanding and requires experiencing death as a sensory reality: To feel (the coldness of the body),To see (the change in skin color),To smell (the absence of the baby’s own scent).

If the body is sufficiently cooled and adequately cared for, it will generally change very little in the first few days. If the parents are offered repeated opportunities to be close to their dead child, the reality of its death reaches them at every level of understanding, thereby opening the path to a healthy grief process.

As described in this article, the primary focus of support and counseling within the context of PnPC lies in extending offers that respect the parent’s fears and emotional defense mechanisms, including their possible decision to refuse an encounter with their deceased child. In resource- and person-centered grief support and counseling, the aim is always to stand by the parents in their decision-making process, giving them the opportunity to expand their scope for creative decision-making by offering gentle and compassionate prompts. Support and counseling should never be a matter of leading a person in one direction in the belief that the counselor knows what is best for the mother or the parents.

### 3.4. R—Room for Bonding

Parents need time, opportunities, and encouragement to grow into their roles as parents. The establishment of an emotional bond with a dying or deceased baby must develop under challenging circumstances. Here many parents need professional encouragement and support. As noted above, many parents initially hesitate to form an emotional attachment to their children. They are consequently dependent upon the advice of people who can explain how important it is not to repress critical life events but rather to assign a temporal, emotional, and spatial place for them in the biography of one’s life so that they may be successfully integrated and not remain as unresolved emotional trauma [14]. For these reasons, it is especially important to respectfully encourage parents to build a bond with their children without pressuring them to do so.

The parent-child bond can be further strengthened, even after death. This strikes many bereaved parents initially as being impossible to happen and perhaps impossible to bear. 


**
*Quotation of a parent*
**
*: “We would like to remember him as he was when we left him on the unit”.*


The thought of revisiting their dead child is initially daunting to many parents. It takes time, and the reassurance of the support of a counselor, for the parents to reconcile themselves with the idea. Gentle and compassionate prompting, as well as sharing the experiences of other parents in the same situation, may allow the parents to intellectually embrace the idea as they make their own affirmative decision. 

Experience reveals that many parents initially reject the idea of revisiting their dead child. Many harbor unspoken and often inaccurate mental images about how much their baby may have changed after death. It is important at this point to inquire specifically into the reason for their rejection in order to be able to gently replace imagined misinformation with factual insight. Should the parents persist in not-wishing to revisit their dead child, their decision, of course, should be accepted without question. Not uncommonly, however, during the subsequent days, a growing longing to see their child again may develop, even in parents who, at the moment, insist that such a thing is unimaginable. 

It is helpful for the parents to realize that many other parents feel this same reluctance, but it is also important to ensure that they very clearly understand if and for how long a time the opportunity to see their child again will be available. The same parents who at first hesitate to revisit their dead child not uncommonly seek to repeat the encounter several additional times. During these visits, they have the opportunity to act like parents and to do the things that parents do. This might include washing, diapering, and dressing their child. Parents might like to take their children for a walk outside or play a lullaby on the guitar. Through the act of doing, a connection is established that can further support the bonding process even after the death of their child. 

The bond established with their baby also allows the parents to define themselves as *parents*, potentially providing them with opportunities for creative and active decision-making, allowing them license to assign their child a rightful place in their lives. This does not diminish their grief, but it does allow them to grieve openly and unselfconsciously for a child whom they love, miss, and who ultimately belongs to them.

Rather than feeling stigmatized or ashamed of themselves, they can now permit themselves to be proud of their baby. Often, they prefer the company of those people who are most important to them to be present alongside them as they stand in admiration of their dead child. The more their self-image embraces their parental role, the better they can personalize the irretrievable time remaining before the funeral, and the more numerous the memories they will have a chance to internalize—memories, which must last for a lifetime, and which cannot be recouped. 

For many parents, it is helpful to know that their grief will modify over the course of time and that there will again be space for new, positive feelings. Their love for their deceased child, however, will never change. It will remain an integral part of the rest of their lives. Parents should ideally not be left with any regrets regarding some action left undone because no one at the time made them aware or encouraged them to act. These are things that bereaved parents cannot be expected to know in advance and may therefore require a foresighted accompaniment offering anticipatory support and guidance [10].

### 3.5. O—Owning a New Life-Situation

Bereaved parents are, and always will be, parents. The dead child, no matter how young or small it was, nor how brief its lifetime, remains a unique and irreplaceable person. When a child dies around the time of its birth, many parents question if they are now still a mother or a father. They often wrestle with how to answer when they need to officially stipulate how many children they have. Particularly for parents who had been expecting their first child, its death often brings into question their respective roles. They are a (married) couple who became parents, only to now have become bereaved parents. This can present a special challenge to their relationship, stressed further by the demands of their separate individual grieving needs [2,15,16,17,18,19].


**
*Quotation of a parent*
**
*: “It seems as if the pregnancy and our son had never existed. On the outside everything was the same as before my pregnancy, but inside there was a storm of despair and I knew things would never be the same again.”*


Additionally, the perception by family and friends that “if you don’t talk about it, it won’t hurt as much” is one that most parents experience as very hurtful and heightens their concern that their little child will soon be forgotten or may be seen as “replaceable”. “You two are still young, you can still have lots of children” is another comment that parents may hear from well-intentioned but helpless relatives. In the process of grief counseling, these common attitudes should be addressed beforehand to prepare parents for such possible situations, specifically to give them the opportunity to formulate a response that works best for them. 

The dying or deceased child needs a firm place in the life biography of its parents and within the larger family circle. Only when the child is given a place within the family does its absence become recognizable to those it left behind. From this arises a recognition of the parents’ loss and an understanding of their grief.

### 3.6. W—Walk Beside and Guide

Transitions herald change. Parents who are about to or have already lost their child find themselves in a highly stressful situation, one which can lead to emotional instability and even long-term illness if appropriate supportive resources cannot be mobilized [6,20,21]. In such stressed situations, the ability to compile resources to adequately cope with changes and to adapt to the new situation is significantly reduced. Each change increases the stress burden for the parents. Among its many goals, grief counseling aims to activate personal and social resources sufficient to ensure emotional stability. 

Parents experience stability in multiple ways: self-efficacy, a sense of control, diversion, connection to empathetic persons, as well as feeling oriented to time and place. Familiar and predictable routines, even if in place for only a brief time, may provide support. They function as a source of strength from which grieving parents can draw as they face upcoming changes. 


**
*Quotation of a parent*
**
*: “Even though it may sound strange to others, I looked forward to every visit with my baby. I knew she was dead, but being there with her, in the farewell room, such peace came over me. I had a purpose; my day had a rhythm. I was visiting my little daughter.”*


Every change in routine threatens to unsettle the stability that has been gained. The mourning process is a dynamic one and prone not only to strong shifting feelings but also to unavoidable and not always predictable outward behavioral changes. Most often, however, the parents themselves can sense when some sort of change is imminent. 

Many find it helpful to be accompanied during times of transition insofar as change tends to engender anxiety and uncertainty. Transitions may be many and varied: the moment of diagnosis; the (still)birth of the child; the process of dying; the final exit from the unit; ending lactation, the last contact with the deceased child; the first contact with the funeral home; the funeral itself; visits to the cemetery; the return to everyday life at home; return to work; the first anniversary of the baby’s birth and death; the subsequent pregnancy. Transitions are not to be underestimated and should elicit offers from as many persons as possible to lend their supportive presence. 


**
*Quotation of a parent*
**
*: “I knew that my little son needed no more breast milk. But it was so hard to stop pumping. Somehow, pumping still made me feel so connected to him. Without my midwife’s patience and understanding, I would have had an even harder time finally deciding to stop pumping.”*


In addition to professionals such as midwives and funeral staff, familiar and patient individuals from the private sphere can take on this important task, as can specially trained grief counselors or self-help groups [6,22,23].

### 3.7. F—Forget-Me-Not

Memories are the “proof” that the dead child once existed [24]. The sooner that parents can be encouraged to begin creating memories, the clearer it will become to them that parenthood and the family time they have available to them are limited. Sometimes the pregnancy itself can already be considered as ‘family time’ when the life expectancy of the child is anticipated to end shortly after birth, and this prenatal period thus takes on a special role. 


**
*Quotation of a parent*
**
*: “And then it suddenly struck me, that my baby would likely live longer in my belly than it would after would after it was born…”*


While many parents at the start of their grieving process need suggestions for ways to preserve their memories of this time, as they become more immersed in the process, they increasingly develop their own creative ideas. This can include the involvement of family members and can lead to their getting to know the child and to experience for whom it is that the parents are grieving. Creating memories supports parents both in their efforts to recognize the unforgettable uniqueness of their baby as well as to document their infant’s existence and their time as parents. 

### 3.8. U—Utilizing Resources and Assessing Risks

Personal grief is a healthy reaction to the loss of a loved one. Grieving parents are, like most people, “experts in their own needs” and essentially possess individual coping strategies. From the moment the diagnosis, which marks the onset of the grieving process [1], parents are put into a shock-like state. Typically, they experience themselves initially as powerless, overwhelmed, paralyzed, disoriented, and deprived of any control over the situation. Reactions to these severe stressors can be very diverse and may fluctuate over the course of time.

Special attention should be paid early on to risks that might complicate or possibly prevent a healthy grieving process. Included among the potentially most significant risk factors are [4,25,26]: Lack of a close, supportive environment (e.g., where nobody knows the child)Special life circumstances that hinder the mourning process (e.g., parental illness)Suppression /deferral of the grief (e.g., in cases where the deceased child is survived by one or more siblings from a multiple pregnancy)Death under traumatic circumstances (e.g., a traffic accident or complications surrounding a home delivery)An ambivalent relationship with the child (e.g., an unwanted pregnancy)Lack of access to support systems (e.g., no bereavement counseling is available)

The counterbalance to these risk factors lies in the personal resources of the bereaved parent. It is important to identify these resources and utilize them at an early stage in the process of grief counseling; if found lacking, it may be necessary to help initiate them. 

In addition to the possible inclusion of significant persons into the process, the utilization of family and social networks can also be considered. Reliable support services and options give parents the means to develop an individual approach that helps them transition from powerlessness to effectiveness and thereby regain some of their lost sense of control. 

While many parents do not have the necessary information regarding regional support systems, they do believe that they might benefit from such a support system in the course of their grieving process. Often, however, they lack the strength to initiate contact on their own. Ongoing outpatient support (e.g., by a midwife [27,28]) should be set up in consultation with the parents at an early stage while still in the hospital setting and should be part of the discharge planning for transition to home. 

The offer of an early resource-oriented bereavement intervention provides support to the parents by identifying their specific needs and mobilizing their resources. It additionally offers the opportunity to identify risk factors for possible aggravated mourning at an early stage and to take appropriate measures.

### 3.9. L—Leveraging Support Systems

Bereavement support is a multi-professional process. The more comprehensive and individualized the support and assistance provided to the parents during their hospital stay, the more imperative it is to establish adequate follow-up support services or to always ensure access to relevant networks [4]. Parental needs vary during the grieving process [25]. They may potentially require persons with different roles to help them focus on changing issues. While transitioning to the outpatient setting, it can be useful to provide the parents with written information anticipating potential future needs. Ideally, this should be performed at a time when the parents have already taken their own first steps on their path of grief and have regained some of their ability to initiate their own decisions and actions [29]. 

Inter-professionality can create synergy—creative solutions and collaborations—that can be presented to the parents. In addition, a multi-professional approach relieves the burden on the individual caregivers and thus ensures long-term sustainability in the care provided.

Table 1 summarizes and illustrates the essential aspects of the “SORROWFUL” model.

## 4. Discussion

Sickness, dying, and death at life’s very beginning cause not only the parents and their families but also the support staff to question life’s meaning. When is a life “lived”; when is a life “fulfilled”? These questions become even more acute the closer that death follows on the heels of birth. 

Bereaved parents of prematurely lost infants often have far fewer support systems and much less time to grow into their parental role, come to know their baby and establish a firm place for that child within the family. Many parents find themselves confronted with the outlook that “the smaller the child, the lesser the grief” and begin to wonder about the “normality” of the deep and persisting sadness they are feeling for their newborn baby. Even when a child has left behind very few tangible traces of ever having once been present, it occupies an irreplaceable and sacrosanct place in the parents’ lives [30]. Bereaved parents of newborn infants frequently need encouragement and resource-mobilizing guidance to enable them and for them to allow themselves to walk their individual path of grief, trusting that although their grief will someday change, the love for their child never will. Grieving styles are as individual as the individuals who grieve. It is, therefore, important in the context of bereavement counseling to perceive the entire personality of each individual mourner, to grasp their needs, and to respect their individuality [31]. 

We hope that the ‘SORROWFUL’ model presented in this paper may offer guidance to those tasked with providing individualized support to bereaved families (i) during pregnancy, birth, and the postnatal period, (ii) in the outpatient and hospital setting, and (iii) at the time of diagnosis of a life-limiting disease until well after the death of the child.

The aim of this model is to raise awareness among all those involved in the care of dying and deceased newborns of the needs presented to them by parents prematurely bereft of their babies. We have sought to present the model in terms that underscore its applicability to different care settings. The intent of this rubric was not to identify a gold standard for grief counseling but rather to encourage counselors to develop their own systems with the resources available to them, broadening the scope of individualized care options for the parents. Grief is not an illness but a natural reaction to loss. Early, professional, and individualized grief counseling can help integrate the crisis experience of a child’s early death into the life biographies of its parents, potentially establishing new connections. Successful integration may not only facilitate a better chance for re-entry into the world of social, professional, and relational living but also carries societal benefits in avoiding protracted treatment costs for secondary diseases associated with prolonged grief disorders or depression. There exist good political justifications for financially supporting an expanded implementation of grief counseling as an initial component of patient or family care within the framework of palliative care.

We recognize that implementing all elements of the ‘SORROWFUL’ model into a PnPC program may be constrained at times by the limitation of resources, staff, and particularly funding, some of which may need to be developed from outside financial sources. Regional solutions and adaptions will necessarily need to be based on available facilities and human resources. 

Despite having been developed in a higher-income European region, we believe that the fundamental elements of the “SORROWFUL” model can be incorporated into PnPC approaches in other cultural and/or lower resource settings. Scenarios similar to those described in this paper will benefit from the practical suggestions offered by the model, a model which at the same time leaves room for cultural and individual adaptations, accommodating the varying needs of affected parents and the resources available.

The “SORROWFUL’ model does not propose to answer all questions regarding psychosocial support within the context of PnPC. It has, however, shown itself at our institution to be a valuable tool to help orient the providers of such care. It should rather serve as orientation, encouraging successful adaptation to similar problems within other cultural and resource settings, and may hopefully prompt national and international discussion about important issues attending perinatal palliative care.

## Figures and Tables

**Table 1 children-10-00038-t001:** The ‘SORROWFUL’ model—fundamental elements of psychological support for parents grieving the (impending) loss of infant(s) during peri- or neonatal life.

**S** teadfast Support	Most families do not seek out the support of counseling services, especially in the acute situation. Proactively contact the affected family (in person, in writing, or by phone). It is not important whether you have already been introduced or not.
**O** ffer Gentle and Compassionate Prompts	Ask the parents if you may share with them the experiences of other families and inquire as to their own thoughts. Thinking aloud together—focused on specific needs—can help unscramble thoughts, put the situation into perspective and suggest an expanded range of options for action.
**R** ealizing and Comprehending Death with all Senses	It helps parents, especially in the beginning, to know that they will not be left alone when meeting their deceased child. Encourage them to revisit their child without pressuring. Offer parents the opportunity to meet their dying/deceased child again and again.
**R** oom for Bonding	Allow parents to meet and get to know their child so they can genuinely say goodbye. Attachment to the child does not intensify grief but rather supports a healthy grieving process. Give parents time to do important parenting activities that will continue to support the bonding process even after death.
**O** wning a New Life-Situation	Giving the child a place in the family means being able to sense the absence that he or she leaves behind after death, even if family members and friends have never even met him or her. Encourage parents to invite the people who are significant in their lives to meet their dying/deceased child, so they will know for whom it is that the parents are grieving. Experience shows that they who have met the child will better understand the parents’ sense of loss.
**W** alk Beside & Guide	Provide parents with a fully informed sense of how grieving may vary from one person to another. Provide the information gently but also repeatedly because parents usually do not remember fully if they have been told only once. Ensure safety by ensuring that parents having ready access to information and/or support services when important changes may be imminent.
**F** orget-Me-Not	Memories and mementos attest to the life and existence of the dying/deceased child. Encourage and support the family (and their creativity) during the irretrievable time until the funeral to collect as many memories as possible.
**U **tilizing Resources and Assessing Risks	Gaining the parents’ trust takes time. Offer them support as early as possible.Early use of resource-oriented support services may reduce the risk of an excessive or dysfunctional grieving experience.
**L** everaging Support Systems	The individual needs of parents change during the mourning process. They require a range of support persons possessing an open mind and a readiness to engage in multi-professional support and interprofessional cooperation.Multi-professional support expands the parents’ range of options as well as increases awareness of their own needs and capabilities.

## Data Availability

Not applicable.

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
