# Peer review of "Psychosocial Support within the Context of Perinatal Palliative Care: The “SORROWFUL” Model†"

_children, 2022, doi:10.3390/children10010038_

Round 1

Reviewer 1 Report

The authors present a model for supporting parents of children who die close to birth. The model is based on their professional experience and past research. The SORROWFUL model includes Steadfast support, Offer gentle and compassionate prompts, Realizing and comprehending death with all senses, Room for bonding, Owning a new life situation, Walk beside and guide, Forget-me-not, Utilizing resources and assessing risks, Leveraging support systems. The authors provide a compelling description of the challenges facing parents of children who die in the neonatal period and what kinds of support they need. The authors appear to be describing an ideal approach based on their experience. While most of the recommendations are not particularly novel (with some exceptions noted below), the authors present a compelling case for their approach and highlight some issues that are specific to neonatal bereavement.

 As the special issue is about international approaches to palliative care, the authors may want to discuss whether any aspects of this model are more specific to Germany/Europe and which aspects of this model might be difficult to implement in other countries and health systems with limited resources.

The authors appear to be describing the practices in their own neonatal unit. I wonder if the authors could provide more information about the unit, what kinds of patients they usually see, and how they were able to offer some of these services such as psychosocial support.

I liked the quotes from parents under each heading, but the authors may wish to say where these quotes came from (e.g. are they parents in their program, were interviews conducted for another study?)

On lines 63-68, The authors may wish to move the citation for Li et al. 2003 so that it follows the first sentence in the paragraph.

The authors make excellent points about how others may assume that the very short life of the child means the parents do not suffer as much and the parents may therefore get less support or acknowledgement from others. I particularly appreciated the point on line 78-79 that part of why the relationship between the family and the neonatal staff is so important is because the staff are among the few people who meet and interact with the baby.

Section 2.1 makes an important argument for why actively reaching out to support parents is important versus waiting for parents to contact the team if they need anything. Advocating for proactive counseling service which makes sense. The authors may wish to acknowledge that this may be difficult for some programs to implement when resources and staff are already stretched thin and outside funding may be needed for some programs.

Sections 2.3 and 2.4 are the most novel parts of this paper. The authors make compelling arguments for allowing parents time to bond with their child even after death. The authors may wish to mention that, in addition to individual preferences,  clinicians may also need to be familiar with cultural differences in how individuals interact with the deceased. Some parents think it is inappropriate to be in the same room as their child is dying based on their tradition. Others may have loud, emotional displays of grief after the child dies which are normal in their culture but may be unsettling to staff and other families on the unit.

The authors may also wish to acknowledge that their suggested approach of allowing parents to visit over multiple days may be unusual compared to other programs and difficult for some programs to implement. In the United States I think most programs try to give parents a few hours in a quiet room after the child dies before the body is taken to a funeral home. I think it would be logistically challenging to arrange space and staff for multiple visits over several days. The authors may wish to say more about whether this approach is unique to their institution or whether it is used at other institutions or other countries.

The authors may also wish to say more about how they were able to implement these practices in their unit and how they were able to arrange for parents to do things like take the child for a walk outside (e.g. does this take place in a courtyard or garden attached to the hospital).

The authors may also wish to say more about whether there was any resistance from staff or administration to this approach. Did it take time to convince staff that parents would benefit from this approach?

Reviewer 2 Report

The paper reports on a newly developed model of support for parents who have experienced the death of a newborn child. Impact of such a bereavement on the parents can be very harsh and the introduction describes the psychosocial aspects of the grief of the parents, and importantly, also notes the role of clinical staff.

The model presented in this paper is very interesting. It is described in a comprehensive way, and I have the impression that authors have given it a lot of thought. However, it is not clear how the model was developed. If could help the reader if authors would include a ‘methods’ section in the manuscript and describe the procedure and methodology used for developing the model.

In the same vein, authors should explain where the quotes are coming from, and what their purpose is. Without context, these are a bit anecdotal.

At the end of the manuscript authors stated that they hope that the model will be useful. It would strengthen the manuscript if authors could explain how the model will be used. How will it be implemented and evaluated. If not, it remains a model like any other. Please explain the next steps. What are you going to do with it, or how can others use this model? What are the implications for practice, research, policy?

Round 2

Reviewer 1 Report

The authors present a model for supporting parents of children who die close to birth. The model is based on their professional experience and past research. The SORROWFUL model includes Steadfast support, Offer gentle and compassionate prompts, Realizing and comprehending death with all senses, Room for bonding, Owning a new life situation, Walk beside and guide, Forget-me-not, Utilizing resources and assessing risks, Leveraging support systems. The authors provide a compelling description of the challenges facing parents of children who die in the neonatal period and what kinds of support they need.  The authors present a compelling case for their approach and highlight some issues that are specific to neonatal bereavement.

The authors have responded to the concerns I expressed in my prior review by adding more detail about their program and how they developed their model. They have also provided more detail about how their model could be applied in other settings and acknowledged the international variability in programs and the challenges of financially supporting the approach they describe.